# Updating the Mechanism of Bicarbonate (HCO_3_^−^) Activation of Soluble Adenylyl Cyclase (sAC)

**DOI:** 10.3390/ijms26136401

**Published:** 2025-07-03

**Authors:** Jacob Ferreira, Hayden Belliveau, Clemens Steegborn, Jochen Buck, Lonny R. Levin

**Affiliations:** 1Department of Pharmacology, Weill Cornell Medicine, New York, NY 10065, USA; jaf4002@med.cornell.edu (J.F.); llevin@med.cornell.edu (L.R.L.); 2Department of Biochemistry, University of Bayreuth, 95440 Bayreuth, Germany; clemens.steegborn@uni-bayreuth.de

**Keywords:** soluble adenylyl cyclase, bicarbonate, pH regulation, cyclic AMP

## Abstract

Soluble adenylyl cyclase (sAC) is molecularly and biochemically distinct from other mammalian nucleotidyl cyclases. It is uniquely regulated directly by bicarbonate (HCO_3_^−^) and calcium (Ca^2+^) ions and is responsive to physiologic fluctuations in levels of its substrate, adenosine triphosphate (ATP). Our initial in vitro biochemical studies suggested two mechanisms for HCO_3_^−^-dependent elevation of sAC activity: increasing catalytic rate and relieving inhibition observed in the presence of supraphysiological levels of substrate, ATP. Structural and mutational studies revealed that HCO_3_^−^ increases catalytic rate via the disruption of a salt bridge that facilitates productive interactions with the substrate. Here, we demonstrate that the HCO_3_^−^ stimulation observed under supraphysiological ATP concentrations is due to the mitigation of ATP-dependent acidification. Therefore, we conclude that the sole physiologically relevant mechanism of HCO_3_^−^ regulation of sAC is through its pH-independent effect facilitating productive substrate binding to the catalytic site.

## 1. Introduction

Many intracellular signal transduction pathways are mediated by the ubiquitous second messenger cyclic adenosine monophosphate (cAMP). Adenylyl cyclases (ACs) produce cAMP by cyclizing adenosine triphosphate (ATP). In mammalian cells, there are 10 known AC isoforms encoded by the genes *ADCY1-10*, which can be divided into two classes: transmembrane adenylyl cyclases (tmAC) and soluble adenylyl cyclases (sAC). The nine tmACs (*ACDY1-9*) are regulated by heterotrimeric G proteins and mediate cAMP responses downstream from hormone and neurotransmitter signaling via G protein-coupled receptors [1,2]. Conversely, sAC (*ACDY10*) is not bound to the cell membrane and instead is localized to various intracellular compartments including the cytosol, mitochondria, and nucleus [3,4,5,6,7,8,9]. As with other Class III nucleotidyl cyclases, sAC activity requires the intramolecular dimerization of two related catalytic domains (C1 and C2) [10]. Regulation of sAC occurs directly through HCO_3_^−^ and Ca^2+^ ions [11,12,13], and its activity is sensitive to physiologically relevant ATP fluctuations [11,14].

While Ca^2+^ ions stimulate sAC activity by increasing the apparent affinity for the substrate ATP [11], HCO_3_^−^ stimulates both Mg^2+^-dependent and Mg^2+^/Ca^2+^-dependent cyclase activity with an EC_50_ within the physiologic intracellular range of HCO_3_^−^ in cells (i.e., 10–25 mM) [11,13]. Initial in vitro cyclase assays on purified sAC protein suggested that HCO_3_^−^ activation occurred via direct binding and was independent of changes in pH [13]. Subsequent kinetic studies predicted two mechanisms for HCO_3_^−^ activation of sAC activity, increasing max velocity (V_max_) and relieving a putative substrate inhibition observed under conditions with supraphysiological levels of ATP [11]. Structural, mutational, and biochemical studies of human sAC identified the bicarbonate binding site (BBS) and revealed the molecular mechanism for the HCO_3_^−^-dependent elevation of the enzyme’s k_cat_ [10,15,16]. The BBS is adjacent to the ATP-binding catalytic site, and both sites include residues from C1 and C2. Two important residues in the BBS are Lys95 and Arg176. In apo human sAC, a salt bridge is formed between Arg176 and a residue essential for ATP conversion, Asp99. The Arg176–Asp99 salt bridge creates an inhibitory interaction. HCO_3_^−^ binding engages Arg176, breaking the salt bridge and releasing Asp99, which allows it to flip towards the active site, facilitating ion site formation and ATP binding and turnover. Arg176 acts as a “switch” connecting the active site and the BBS, meaning its movement dictates whether the enzyme is in the active or inactive conformation. Thus, direct binding of HCO_3_^−^ to the BBS favors productive ATP binding elevating the enzyme’s V_max_ [10,15].

In contrast, structural and kinetic studies provided no insight into the molecular basis for the substrate inhibition observed at supraphysiological ATP levels (>7.5 mM) and the hypothesized second HCO_3_^−^-dependent activation mechanism via mitigation of this inhibitory effect. Here, we investigate this second mechanism of HCO_3_^−^-dependent stimulation of sAC in vitro activity.

## 2. Results

Our previous studies exploring sAC sensitivity to pH focused exclusively on the effects of basic pH and its possible role in HCO_3_^−^-induced sAC activation [13]. Here, we expanded these studies to ask how sAC in vitro activity is affected by acidic pH. When assayed at a physiologic or slightly basic pH, the cyclase activity of sAC is unchanged. However, as the pH becomes more acidic, cyclase activity decreases until there is almost no measurable cyclase activity at a pH of 5.5 (Figure 1). This acid sensitivity is consistent with the catalytic roles of several acidic residues [10,15]. With the appreciation that the cyclase activity of sAC is highly acid-sensitive, we revisited whether the previously used assay conditions supplied sufficient buffering capacity to neutralize the pH of solutions containing supraphysiological levels of ATP, whose disodium salt (the form used in all our assays) is a weak polyacid. Our previous kinetic experiments were performed in the presence of 50 mM of Tris at pH 7.5 [11]. The pH of 10 mM of disodium ATP dissolved in water in the presence of 50 mM of Tris proved to be acidic, at ~6.1. In contrast, 100 mM of Tris (pH 7.5) was adequate to buffer 10 mM of disodium ATP to remain at ~7.2 (Figure 2). Interestingly, an addition of 40 mM of HCO_3_^−^, a concentration which had successfully mitigated the previously observed substrate inhibition [11], neutralized the pH of 10 mM of ATP solution in 50 mM of Tris to ~6.8 (Figure 2). These data suggested that the substrate inhibition observed at high levels of ATP, which was mitigated by HCO_3_^−^ in previous experiments, may be due to inadequate buffering by 50 mM of Tris.

We tested this hypothesis by measuring cyclase activity of sAC in the presence of two concentrations of ATP (2 and 10 mM), two concentrations of Tris (50 and 100 mM), and in the presence and absence of HCO_3_^−^. At physiological ATP levels (2 mM), there was no significant difference between the activity observed in the presence of 50 mM of Tris relative to 100 mM of Tris, and at both pHs, HCO_3_^−^ stimulated sAC activity to a similar degree (Figure 3A). In contrast, when measured in the presence of supraphysiological levels of ATP (10 mM), there was a stark difference between the assays buffered with 50 mM of Tris, where the pH decreased to ~6.1, compared with the assays buffered with 100 mM of Tris, which remained at pH ~7.5. In the assay with 50 mM of Tris, the acidity caused by high levels of ATP reduced sAC activity considerably (Figure 3B). However, when sAC activity was measured in the presence of 10 mM of ATP and 40 mM of HCO_3_^−^, which mitigated the acidifying effects of 10 mM of ATP in 50 mM of Tris (Figure 2), the activity was indistinguishable between 50 mM and 100 mM of Tris (Figure 3B).

Finally, we directly compared ATP kinetics in 50 mM of Tris (pH 7.5) with that in 150 mM of Tris (pH 7.5). These experiments were performed at 150 mM of Tris pH 7.5 to ensure adequate buffering for the highest levels of ATP tested (15 mM). Since Ca^2+^ stimulates sAC by decreasing the enzyme’s K_M_ for substrate ATP, Ca^2+^ and HCO_3_^−^ work synergistically [11]. In the absence of HCO_3_^−^ (i.e., Mg^2+^/Ca^2+^ ATP alone), we observed inhibition at high ATP levels (>5 mM) only in the reactions buffered by 50 mM of Tris; in the reactions buffered by 150 mM of Tris, the K_M_ for ATP was ~1 mM, as previously reported, and importantly, there was no evidence of substrate inhibition even at the highest ATP concentrations tested (Figure 4A). In contrast, in the presence of 40 mM of HCO_3_^−^, there was no discernable difference in the kinetics measured in the presence of 50 mM of Tris compared with 150 mM of Tris. In both cases, the measured K_M_ was between 1 and 2 mM (Figure 4B), consistent with previous reports [11]. These observations reveal that the substrate inhibition observed in 50 mM of Tris in the absence of HCO_3_^−^ is due to the acid sensitivity of the enzyme, and the previously hypothesized HCO_3_^−^-dependent stimulation of activity at supraphysiological levels of ATP was merely due to its ability to mitigate the acidification due to the insufficient buffering capacity of 50 mM of Tris.

In conclusion, the data presented in this paper reveal that the buffering capacity used previously was insufficient at very high levels of ATP and updates the mechanism for HCO_3_^−^ activation of sAC. The previous hypothesized ATP inhibition was in fact due to a decrease in cyclase activity at pH values less than 6.5. Therefore, the only known mechanism for HCO_3_^−^ activation of sAC is through direct binding to the enzyme, which breaks an inhibitory salt bridge (Figure 5) [10,15]. Moreover, moving forward, the optimal conditions for ATP kinetics of sAC to properly control pH are higher concentrations of Tris or ATP stock solutions with pre-adjusted pH to avoid pH shifts in the assay solution.

## 3. Materials and Methods

### 3.1. Materials

Magnesium chloride (MgCl_2_), calcium chloride (CaCl_2_), manganese (II) chloride (MnCl_2_), sodium bicarbonate (NaHCO_3_), dithiothreitol (DTT), bovine serum albumin (BSA), disodium ATP (non-radioactive), cAMP (non-radioactive), Tris-HCl, MES, and MOPS were all purchased from Sigma-Aldrich (St. Louis, MO, USA). [α-^32^P]-ATP and [2,8-^3^H]-cAMP were purchased from PerkinElmer (Waltham, MA, USA).

Heterologously expressed and purified human truncated sAC (sAC_t_) protein was used for all biochemical assays. The sAC_t_ protein was N-terminal tagged with GST and purified from Sf9 cells via baculovirus expression [11].

### 3.2. In Vitro Adenylyl Cyclase Activity Assay

All in vitro adenylyl cyclase activity assays utilized the classical “two-column” method, developed by Salomon [17]. In this assay, conversion of [α-^32^P]-ATP to [^32^P]-cAMP is quantified by sequential Dowex and Alumina chromatography resins to purify the generated [^32^P]-cAMP. Cyclase assays were performed in 100 μL of total reaction volume using 25–600 ng of sAC_t_, 50–150 mM of Tris-HCl (pH 7.5), 3 mM of DTT, BSA 0.03%, substrate [α-^32^P]-ATP, and either MnCl_2_, or MgCl_2_, and/or CaCl_2_, and/or NaHCO_3_ concentrations as indicated. Reactions were initiated by the addition of 10,000,000–1,000,000 counts/minute of [α-^32^P]-ATP. Reactions were incubated for 30 min at 30 °C, quenched by adding 200 μL of SDS 2%, and cAMP was quantified following sequential Dowex and Alumina chromatography. An internal standard of ~10,000 counts/minute of [^3^H]-cAMP was added to each reaction.

For Figure 1, MES or MOPS buffer was used instead of Tris to maintain the pH of the reaction mixtures at pH levels other than 7.5. For ATP kinetics curves (Figure 4), [α-^32^P]-ATP was diluted in 2/3 steps, starting at 15 mM, thus keeping the specific activity equivalent in every reaction.

### 3.3. Statistical Analysis

Plotting data, curve fitting, and statistical analyses were performed using the GraphPad Prism software (GraphPad, version 10.2.3, San Diego, CA, USA). All data are shown as the mean ± SD or SEM.

## Figures and Tables

**Figure 1 ijms-26-06401-f001:**
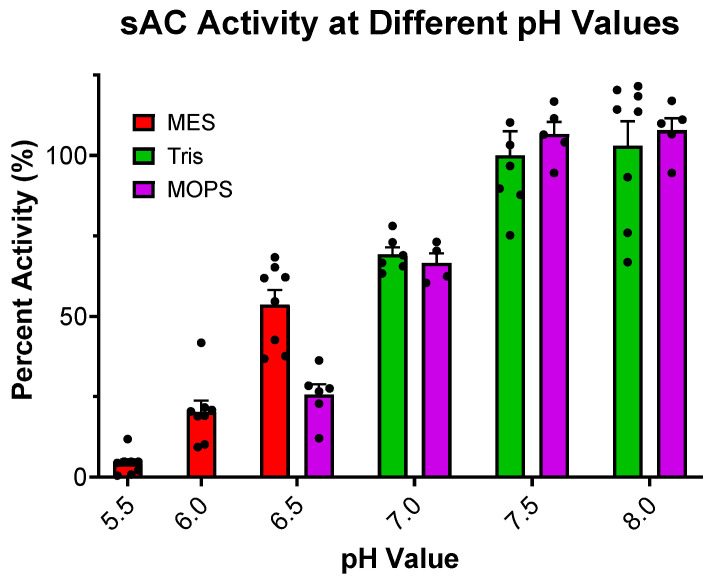
Effect of pH on sAC cyclase activity: In vitro cyclase activity assay of recombinant sAC_t_, in the presence of 10 mM of Mn^2+^, 2 mM of ATP, and 50 mM of respective buffer to maintain the pH indicated. Mn^2+^ was used in this experiment for full catalytic activity. All data were normalized to activity at pH 7.5, the standard pH used previously.. Bars represent averages, with each dot representing individual replicates and errors bars indicating standard errors of the means. The addition of 75 mM of MES or 75 mM of MOPS had no effect on sAC activity when tested in the presence of 75 mM of Tris at pH 7.5 (Appendix A).

**Figure 2 ijms-26-06401-f002:**
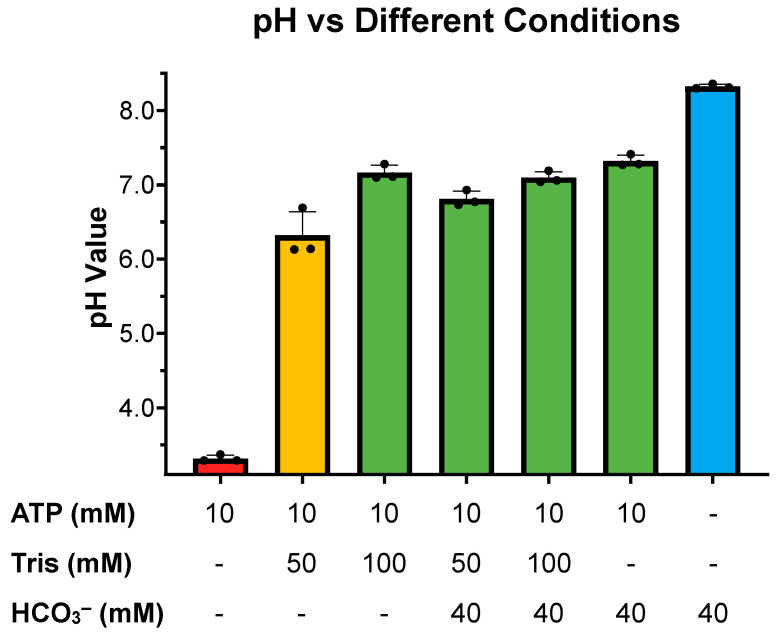
pH of different cyclase reaction components: Final pH of different concentrations of Tris pH 7.5, ATP, and HCO_3_^−^ as indicated. Bars represent the averages of 3 independent pH measurements taken at room temperature (~22 °C) with each dot representing individual replicates and error bars indicating standard error of the mean.

**Figure 3 ijms-26-06401-f003:**
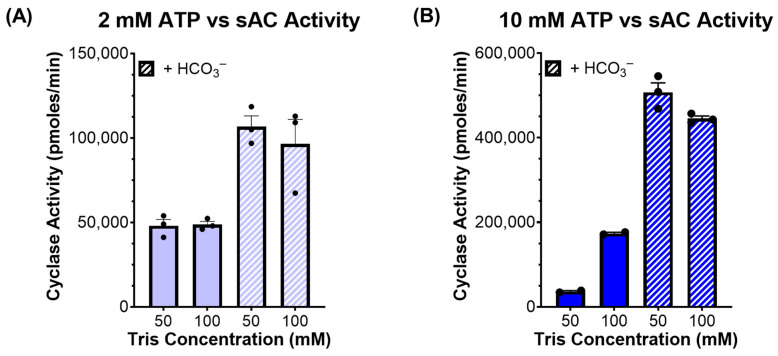
sAC activity in the presence of different cyclase reaction components: In vitro cyclase activity assay of recombinant sAC_t_, in the presence of Mg^2+^ and Ca^2+^, the absence and presence of 40 mM of HCO_3_^−^, the presence of either 50 mM or 100 mM of Tris, and either (**A**) 2 mM of ATP or (**B**) 10 mM of ATP. Bars represent the average of triplicate determinations with each dot representing individual replicates and error bars indicating standard error of the mean, from a representative experiment which was repeated 3 independent times.

**Figure 4 ijms-26-06401-f004:**
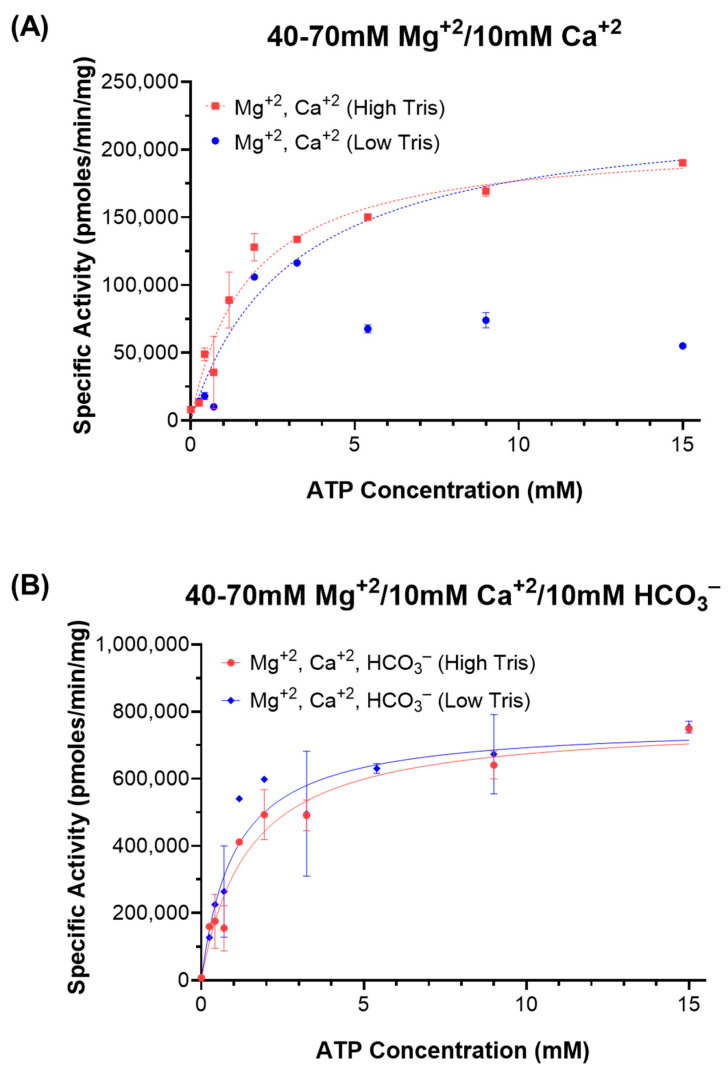
ATP kinetics of sAC in different Tris concentrations: In vitro cyclase activity of recombinant sAC_t_, in the presence of Mg^2+^, Ca^2+^, HCO_3_^−^, and either 50 mM (blue lines; “Low Tris”) or 150 mM (red lines; “High Tris”) of Tris (pH 7.5). Activity was assayed as a function of substrate [Mg^2+^-ATP] in the presence of either (**A**) 40–70 mM of Mg^2+^ and 10 mM of Ca^2+^ (dotted lines) or (**B**) 40–70 mM of Mg^2+^, 10 mM of Ca^2+^, and 40 mM of HCO_3_^−^ (solid lines). Data points represent averages of duplicate determinations with standard error of the mean indicated, from a representative experiment which was repeated 3 independent times. The dotted blue curve was generated excluding the inhibited points at higher values of ATP.

**Figure 5 ijms-26-06401-f005:**
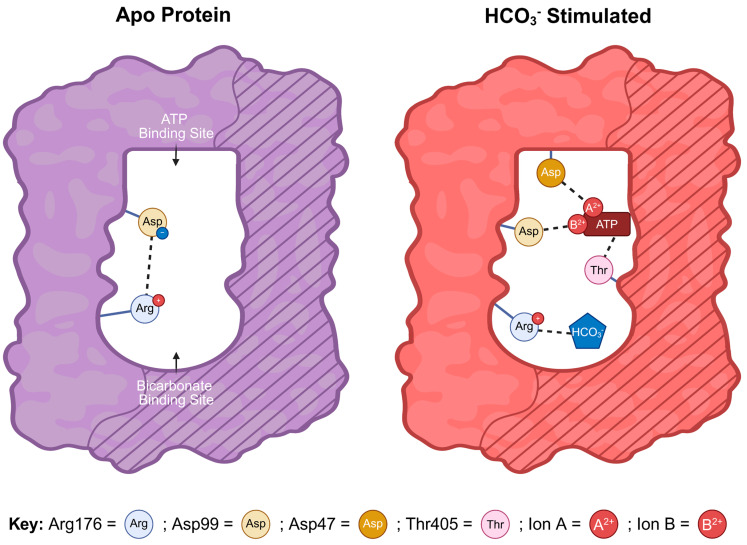
Current model for HCO_3_^–^ activation of sAC: Cartoon representation of the mechanism of HCO_3_^—^-dependent sAC activation. The bicarbonate binding site (BBS) (curved area) and the ATP-binding active site (rectangular area) are formed from the interface of the two related catalytic domains C1 (unshaded) and C2 (striped). In apo human sAC, a salt bridge forms between Arg176 and Asp99, which prevents productive ATP binding. When HCO_3_^—^ occupies the BBS, it interacts with Arg176 breaking the salt bridge and releasing Asp99. Once released, Asp99 is able to reorient towards the active site facilitating ATP binding and catalysis. Thus, Arg176 acts as a “switch” connecting the active site and the BBS. In the active complex, residues from both catalytic domains participate in ATP binding; Asp99 and Asp47 from C1 each interact with ATP via divalent cations (i.e., Mg^2+^, Ca^2+^) and Thr405 from C2 interacts with the adenine group of ATP [10,15]. A previously proposed mechanism whereby HCO_3_^—^ relieved ATP inhibition was due to acid sensitivity.

## Data Availability

Data is contained within the article and Appendix A.

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
