# Peer review of "Updating the Mechanism of Bicarbonate (HCO3−) Activation of Soluble Adenylyl Cyclase (sAC)"

_ijms, 2025, doi:10.3390/ijms26136401_

Round 1
Reviewer 1 Report
Comments and Suggestions for Authors
Soluble adenylyl cyclase (sAC) is an important component of intracellular cAMP signaling and associated physiological functions. In this study, Ferreira et al. revisit their previously proposed two mechanisms of activation of sAC by HCO₃⁻. The authors find that the previously observed disinhibition of sAC by HCO₃⁻ was mainly a consequence of poor buffering conditions in the presence of supraphysiological ATP concentrations. The higher concentration of ATP (2–10 mM), in the absence of sufficient buffering, lowers pH via its acidic properties and inhibits the enzyme activity of sAC. Addition of HCO₃⁻ can mitigate this ATP-dependent acidification and thereby improve sAC activity. This study will be not only important for its update on the mechanism of regulation of sAC, but also provides an important case study of “how important it is to evaluate physicochemical properties of molecules such as nucleotide triphosphates and ensure good buffering in invitro experimental conditions.”
While I support the publication of this manuscript, I have some major concerns that must be addressed to consider the manuscript suitable for publication.
Some of the major concerns are as follows:
As the buffering capacity of various buffer systems is limited to certain pH ranges, the authors should have ideally performed pH titration experiments using a cocktail buffer system. For instance, mixing equimolar concentrations of MES, MOPS, and TRIS and adjusting the pH to the desired range. This would greatly improve the enhance experimental robustness by providing uniform buffer conditions across the pH spectrum. Some of the issues arising from current experimental design can be seen in Supplementary Figure 1. Here according to the figure labels, sAC in the presence of 50 mM MES at pH 5.0 is as active as at pH 7.5. This is confusing and inconsistent with the authors’ claim that sAC is sensitive to acidic pH conditions. Did authors adjust pH after 50 mM MES or MOPS buffers to 150 mM Tris at pH 7.5–8.0? Since the final salt concentrations differ across conditions, attributing changes in enzyme activity solely to pH is problematic. I strongly recommend that these experiments be repeated with improved buffer standardization, which would significantly strengthen the conclusions.
The authors should also generate a scheme to highlight the proposed changes to the mechanism of activation of sAC by HCO₃⁻.
Minor issues:
Since the pH of Tris-HCl buffers is temperature-dependent, the temperature at which pH was measured should be explicitly stated for all relevant experiments (e.g., in Figure 2).
The data points of all three independent experimental repeats should be depicted in the respective graphs. The statistical significance of the observed activity changes should be indicated, including the mode of error measurement (SEM or SD). This should be mentioned in the figure legend of each plot for data transparency.
Line 72: The statement “pH of solutions containing supraphysiological levels of ATP, which is an acid” is technically correct but sounds unclear or controversial. It would be helpful to clarify this in biochemical terms (e.g., “ATP is a weak polyacid”)
Reviewer 2 Report
Comments and Suggestions for Authors
Major comments:
I understand the significance of the study; however, the proposed title, “Further Insights into Mechanisms of Bicarbonate (HCO₃⁻) Activation of Soluble Adenylyl Cyclase (sAC),” is not appropriate, as it reflects a correction of their previous error rather than a new discovery.
Bicarbonate ion (HCO₃⁻) in an aqueous medium produces carbon dioxide (CO₂) and hydroxide ion (OH⁻). Under atmospheric conditions (i.e., low CO₂ pressure), CO₂ in the medium will evaporate, which is known to raise the pH. Similarly, the binding of divalent cations (such as Mg²⁺) to ATP can lead to acidification of the medium by releasing protons from the phosphate groups of ATP. Since the authors acknowledged that their previous experiments were not carefully performed under controlled pH conditions, I wonder whether the pH was adequately controlled in the presence of bicarbonate in all their earlier studies. This seems like a good opportunity to revisit the pH conditions in all their experiments, ensuring that the pH of every medium is strictly regulated.
Minor comments:
It is important to include detailed information about adenosine triphosphate (ATP), as it is commercially available in different forms such as disodium salt, dipotassium salt, Tris salt, and magnesium salt. Each form may result in a different pH in the stock solution.
Round 2
Reviewer 1 Report
Comments and Suggestions for Authors
The authors have adequately addressed the previous comments and made the necessary changes. I recommend accepting the revised manuscript for publication.